# Patterns of Agricultural Diversification in China and Its Policy Implications for Agricultural Modernization

**DOI:** 10.3390/ijerph18094978

**Published:** 2021-05-07

**Authors:** Hongyun Han, Hui Lin

**Affiliations:** China Academy for Rural Development, School of Public Administration, Zhejiang University, Hangzhou 310058, China; linmumuj@126.com

**Keywords:** agricultural diversification, policy implication, agricultural modernization

## Abstract

Based on the value of agricultural farm products produced by different subsectors in China, the foregoing analysis reveals the dynamic character of agricultural diversification by which, this study seeks to examine the evolutionary process of Chinese agriculture through a quantitative study of agricultural diversification at both national and provincial levels. In the initial stages of reform and opening up, the degree of agricultural diversification in the southwest was relatively high; then the center of agricultural diversification gradually moved to the southeast of China; finally, the degree of agricultural diversification in the economically developed eastern provinces was obviously higher than those in other regions in 2019. It was seen that some provinces in the eastern and central south regions moved toward increasing diversification in one direction, and other provinces changed direction, first moving toward diversification and later toward concentration or vice versa. These oscillations implied that there was a cyclic tendency of agricultural diversification along with an increase in per capita GDP. Generally speaking, the patterns of diversification differed across regions due to diversified agricultural subsectors resulting from different natural and socio-economic circumstances. In particular, in less developed regions with lower agricultural diversification levels, farming agriculture persistently dominated the leading position with relatively more resistance to modernizing trends in other aspects of agriculture. It is urgent for the Chinese government to figure out ways off reconciling agricultural productivity with environmental quality through the ecological intensification of agriculture.

## 1. Introduction

The strategies for economic growth can be broadly grouped into two classifications: one is specialization, the other is scale or scope economy. Agricultural diversification(AD) has been proposed as an augmented strategy to make use of the economies of scope and scale [1,2], improve regional agricultural development and conserve the environment [3]; another aspect of agricultural development in industrialized societies has been agricultural specialization [4]. On the contrary to diversification, specialized agriculture might lead to trade-offs between the economic achievement and ecosystem function of agricultural systems [5]. Along with steady decline of farm numbers in France, the remaining farms have become more specialized due to the economies of agglomeration [6]; the majority of farms in Europe were mixed farms during the 1950s and 1960s [7,8]; in more recent times, however, mixed farming is not commonly seen in Europe [6]. It is found that there are different extents and patterns of AD among countries due to varying technologies, agricultural support systems and government policies [9]. Policy makers in both developed and developing countries are bewildered by the trade-offs between the conservation of environmental diversity and agricultural development [10].

Although some Asian nations, such as Japan, Thailand, and South Korea, have succeeded in the pursuit of agricultural diversification [11], China did not begin its slow march of AD till the early 1980s [12] with varying processes due to large variations in agricultural endowments and varying development across regions [9]. Despite its falling share in national income, the three issues of agriculture, rural development and peasants have been the core of Document No. 1 since 1982. To attain high food self-sufficiency, since the 1990s agriculture in China has undergone the dramatic shift from a highly diverse and resource-recycling approach to a specialized, high-external input, resource-intensive, commercialized model [13]. In 2007, agricultural modernization, typically characterized as “specialization, intensification and scale enlargement” [6], was first put forward by Document No. 1 in China as a feasible option for smallholder households to increase income [14]; increased specialization and scales can endanger ecosystem functions at broader spatial scales [5]. Constrained by infeasible technologies [15,16,17], non-specialized production is still the main mode in China’s agriculture [18]. The policy priority of tackling food security in China [19] intensified agriculture, which has led to environmental deterioration arising from overuse of chemical inputs [20,21,22]. It is imperative for policymakers in China to balance their focus on the efficient production of agricultural modernization, the maintenance of ecosystems, and prosperous rural countryside pursued by AD.

What are the dynamic trends of AD in the pursuit of agricultural modernization in China? The existing studies have mainly focused on the roles of the diversification in technology, market rationale and risk reducing strategies [23]. To the authors’ knowledge, no quantitative measurement of diversification has been applied to the study of China’s structural change and development [9], which is a basis for effective policy formation in restructuring China’s economy to address the dilemma of economic development while protecting the environment. Balanced development is urgently required for rural revitalization in China. More importantly, the impact of agricultural performance is not confined to China, but also has profound impacts on the rest of the world due to China’s size and continued transitional mode of economic growth [24]. It is believed that China is challenging the Earth’s environment [25] and deeply affects “global patterns of resource production and consumption and their associated environmental and geographical impacts” [26] (p.249). China’s fast growth and special paths of transition have puzzled scholars as a growth miracle [27]. Although the trend of the Chinese production diversification has been reported by some descriptive studies [28,29], few researchers have attempted to quantitatively measure AD associated with structural change [9], and little effort has been devoted to the study on the degree and patterns of diversification across regions [30].

In particular, studies concerning AD at the meso level are limited [3]. This study aims to fill this this space by empirically examining the extent of regional levels and national AD levels. China is a good study area as it is a large and diverse country. Confronted with issues of changing consumer preferences, drastic policy revisions shifting focuses from agricultural production to environmental conservation, and food security, the agricultural sector there has undergone major changes during this era. The purposes of this study are threefold: the study aims to (i) examine the nature of AD across China; (ii) to identify patterns of AD associated with economic growth; and (iii) to summarize the implications of the sustainable use of natural resources and agricultural modernization. Firstly, using a macro-NBS data at district level, spatiotemporal patterns of AD at the district and national levels of aggregation from 1978 through 2019 are outlined. Secondly, this study will provide important factual information for China’s agricultural modernization process by examining in depth the drivers of, and constraints to, the process, especially through structural analysis, answering how agricultural production differs across regions within China and the implications for policy design in the process of agricultural modernization.

## 2. Trends of Agricultural Diversification against Structural Transformation

### 2.1. A Synthesis Analysis on the Measure of Firm and Agricultural Diversification

An increasing effort has been devoted to measuring firm diversification [31], though without a "generally accepted definition" of diversification [32] (p. 9). One variant is to numerically count the number of businesses [33,34]. A comprehensive continuous index of diversity has been proposed as follows (Equation (1)):(1)$$d=1-\sum_{j=1}^{n}p_{j}u_{j}$$
where *p_j_* denotes the ratio of *j*th business in a firm, *u_j_* is the assigned weight, and *n* is named as the number of businesses in the firm [34,35].

Gort (1962) [36] is generally credited with the first quantitative examination of diversification in U.S. manufacturing. Firm diversification is referred as follows (Equation (2)):(2)$$d=\sum_{j=2}^{n}s_{j}$$
where *s_j_* is the proportion of j the secondary product in a firm.

Here *d* is not a satisfactory index because it accounts only for the share distribution between primary products and the aggregate of all secondary products, but does not account for the number of products [37].

Two continuous measures have predominated over a variety of approaches. One, originally developed by Hirschman (1964) [38], weights each business share by itself; the other, proposed by Jacquemin and Berry (1979) [39], weights each *p_i_* by the logarithm of 1/*p_i_*.

Independently proposed by Berry (1971) [40] and McVey (1972) [41], the Herfindahl index (HI) has the form ${s_{j}}^{2}$, where *s_j_* is the ratio of the *j*th product to total sales (Equation (3)).
(3)$$\mathrm{HI}=\sum {s_{j}}^{2}$$

Berry (1971) [40] suggests the form of $d=1-\sum s_{j}{}^{2}$,weighted by itself. It covers the properties of the number and distribution [37].

The entropy index (EI) was developed by Jacquemin and Berry (1979) [39]. It is measured as follows (Equation (4)):(4)d=∑j=1nsjln(1sj)
where *s_j_* is the percentage of either the *j*th firm or the *j*th industry, and n is the number of firm activities.

Based on the concept of firm diversification, several measures of AD have been proposed in empirical studies of farm households [42,43], including HI, the ogive index (OI), the entropy index (EI) and the modified entropy index (MEI) [44,45].

The EI is scale-free by giving proportionally less weight to large businesses [39]. EI is given as follows (Equation (5)):(5)$$\mathrm{EI}={\sum_{n}p}_{i}\log(\frac{1}{p_{i}})$$

Coelli and Fleming (2004) [46] use the OI to measure firm/farm level diversification with an equal distribution as a benchmark or norm [47]. OI is given as follows (Equation (6)):(6)$$\mathrm{OI}=\sum_{n}{\left(p_{i}-\frac{1}{n}\right)}^{2}/1/n$$

Like HI, the OI is also a measure of concentration of subsectors (Equation (7)).
(7)$$\mathrm{MEI}={\sum_{n}p}_{i}\log_{n}(\frac{1}{p_{i}})$$ Here, n denotes the number of subsectors.

Each index has its advantages and shortcomings. HI is regarded as an appropriate measure of industry concentration because the squaring of the market shares relaxes the influence of errors due to the frequent lack of precise data on very small firms [48]. Despite its wide usage, the HI gives relatively little weight to rare species [49,50,51]. Proposed by Hackbart and Anderson in 1975 [52], the EI was inappropriate because of its similar shortcomings to OI in the lack of four properties of continuity, symmetry, extremal property and additivity; the principal advantage of the EI is that it is asymptotically normally distributed [47]. However, the EI is more sensitive than the HI to very small firms, the HI is more meaningful as a measure of concentration [37].

### 2.2. The National Trend of Agricultural Diversification in China

To date, the HI is at the forefront of the news reported [53,54]. HI has widely been used in marketing and corporate firm studies in market concentration [55,56,57,58]. It has been used recently to capture crop diversification in agricultural production [59]. Therefore, the HI is the main measure employed in this study.

Farm diversification has not yet been precisely defined till now [60]. “Success in agricultural development often creates the need for a major reorganization of resources both within the agricultural sector and between agriculture and the rest of the economy” [61] (p.41). Sectoral diversification is consistent with specialization at the farm level [62]. AD could be mixed crops on a farm or in a region [63] or an adjustment in production portfolios [64]. The common elements are the reallocation of farm resources into new farm or nonfarm activities [65]. A number of AD are developed in the literature, including the introduction of non-traditional crops/livestock, the adoption of unusual agricultural practices [66,67], or the provision of non-agricultural products and services on-farm [67,68]. The narrow definition of diversification would exclude off-farm employment as a type of diversification [69]. The narrow definition is employed in our study.

There are four levels of diversification: a farm, a sector, a region, and the nation [70]. Sector and region diversification are the core elements of this study. Two objectives will be accomplished in this study. The first is to discuss the spatial variation in agricultural diversification for 1978 through 2019. The second is an extension of the first in that it seeks to develop a model of the stages of diversification and to apply this model to provincial levels.

The district diversification index under subsector *i* and in year *t* is given as follows (Equation (8) and (9)):(8)$$D_{it}=1-H_{it}=1-\sum_{n}(Q_{nit}/{\sum_{k}Q}_{kit}{)}^{2}$$
(9)$$p_{it}=\sum_{n}(Q_{nit}/\sum_{k}Q_{kit}{)}^{2}$$ where *Q_nit_* is output value of subsector *i* in district *n* in year *t*, they are farming, forestry, animal husbandry, and fishery. A small *D_it_* implies that the share of the subsector of concern is concentrated in a few districts.

Another diversification index can be defined as (Equation (10) and (11))
(10)$$D_{nt}=1-H_{nt}=1-\sum_{i}(Q_{nit}/\sum_{j}Q_{njt}{)}^{2}$$
(11)$$p_{it}=\sum_{i}(Q_{nit}/\sum_{j}Q_{njt}{)}^{2}$$

It is a measure of diversification for a particular district, n is the number of subsectors.

Weighing each business share by itself, HI is as follows (Equation (12)):(12)$$\mathrm{HI}={\sum_{n}p}_{i}{}^{2}$$
where n is the number of subsectors, and *p_i_* represents the output proportion of the *i*th subsector in total agricultural output. With an increase in diversification, the HI decreases. The HI ranges from zero, reflecting complete diversification, to one, reflecting complete specialization.

The HI is used to measure sector and regional diversification based on the data comes from the database of China Stock Market and Accounting Research (CSMAR); the missing data is made up from China Statistical Yearbook, China Agricultural Yearbook and Local Statistical Yearbook. The data cover all 30 provincial prefectures from 1978 through 2019. Hainan province was constructed in 1988—there were 30 provincial regions before 1988. There are in all 31 provincial prefectures; Chongqing is embedded in Sichuan province for consistent data availability, then there are 30 provincial regions in our study.

Figure 1 shows that there has been a substantial change of diversifying agriculture. The diversification degree of China’s agriculture has experienced a trend of first increasing and then stabilizing. Nationally, the degree of diversification peaked 0.61 in 2003, then fluctuated slowly to 0.59 in 2019 with a slight reduction in AD (See Figure 1).

Different measures have both strengths and weaknesses [37]. Instead of a single index of HI, a set of five diversification indices have been used in this study to reflect the extent and trends of AD at national level. Other diversity measures employed include HI, OI, EI and MEI, the output of four sectors were used to calculate HI, OI, EI and MEI, and the results are shown in Figure 2.

Generally speaking, the higher the HI score, the more concentrated the output. EI increases with the rising diversification and approaches zero when there is perfect concentration, the upper bound of the index is log n. The OI is also used in the measurement of the concentration of subsectors. The same methodology is used for MEI. The four indicators uniformly reflect that since the reform and opening up, China’s primary sector diversity has experienced a process of increasing first, and then leveling off from the year 2004. Special attention should be paid to the OI, which is a measure of the concentration of subsectors. Before 1995, there were unbalanced contributions of different subsectors in China; this was why the higher OI before 1995; after that, the concentration level declined and leveled off from the year of 2004. A similar situation happened to the indicator of HI; the uneven distribution among subsectors in China had led to the dramatic fluctuation of indicators.

The HI has been proposed as a commonly used indicator of firm diversification [53]. As mentioned above, HI is the measure of AD employed in following section. Nationally, the mean of farming AD by HI was 0.54, minimum was 0.34, maximum is 0.61, and standardized deviation was 0.01. In general, since the reform and opening up, the degree of AD in China has experienced a trend of first increasing and then stabilizing. In the early stages, the degree of AD in the southwest and Inner Mongolia was relatively high, and then the center of AD gradually moved to the southeast.

On average, 18 of 30 provinces had mean AD lower than that of national average of 0.54, and the other 12 provinces had higher means than that of national average. Diversification at the regional level was a part of the evolutionary process of the agricultural subsectors. In the initial stage, the degree of AD was relatively high in the southwest (i.e., Tibet (0.51), Shanghai (0.51), Inner Mongolia (0.50), and Qinghai (0.48)) while their levels of agricultural diversification had shifted to 0.52, 0.62, 0.52, and 0.60, respectively, in 2019; then the center of AD gradually moved to the southeast, especially Fujian (0.66) and Hainan (0.70) in 1993; in 2006, the center of AD shifted further south to the additional three provinces of Zhejiang (0.66), Jiangxi (0.67), Guangdong (0.64). In 2019, Fujian province ranked in first place with 0.70it was followed by Anhui (0.64), Zhejiang (0.63) and Hubei (0.63) Hunan (0.61), and Shandong (0.60). By contrast, in 2019 the provinces with lower indices were as follows: Gansu (0.39), Xijiang (0.41), Shanxi (0.41), Henan (0.46), Guizhou (0.47) (see Figure 3).

As indicated by Figure 4, these four selected representative provinces (Fujian, Hainan, Gansu and Xinjiang) had different patterns of AD. Although Gansu and Xinjiang provinces had the lowest AD indices, Gansu reached the first turning point of 0.47 in 1988, and the second turning point of 0.45 in 1997; the first turning point of 0.44 was reached in 1987 in Xinjiang, the second turning point of 0.44 in 2002, and the third turning point of 0.44 in 2008. Hainan province had the highest AD, and Fujian ranked the second in terms of AD; Xinjiang had the lowest AD after Gansu province. The AD index of Fujian province was 0.38 in 1978, it was lower than that of Xinjiang in 1978 (0.39); however, its AD increased fast with the highest point of 0.70 in 2019; by contrast, Hainan province reached the highest point of 0.71 in 2011, and then fell to 0.64 in 2019.

There was a big gap between the two selected groups. The extent of AD in the economically developed eastern coastal regions of China was obviously higher than those in other regions. What happened to the agricultural structural transformation across regions in China? The next section will examine the patterns of AD along with structural transformation in China, which will provide valuable information for policy formation in the development of agricultural modernization.

### 2.3. Patterns of Agricultural Diversification along with Structural Changes in China

The U-shaped industrial concentration curve is firstly documented by Imbs and Wacziarg (2003) [71]. The production structure initially becomes more diversified with an increase in per capita GDP; and after the threshold income level, the production structure become more and more specialized [71,72]. This inverted U-pattern relationship between diversification and economic growth has been documented in the aggregate economy and within the manufacturing sector; however, it is unclear in terms of the agricultural sector [73]. The U-shaped relationship is partly confirmed by the case of China.

Figure 5 indicates that the level of diversification increased from 0.34 in 1978 to 0.59 in 2002, then peaked 0.61 in 2003; after the turning point, the degree of diversification fluctuated gradually to 0.59 in 2019. Even with fast growing GDP per capita, the AD index remains relatively stable over more than 17 years, instead of a decline in AD. This is different from what has happened in other developed economies, which deserves further attention to provide a detailed analysis of the patterns of AD along with structural changes across regions.

Along with rapid industrialization and urbanization, structural transformation is typically characterized by a decreased contribution of agricultural output and employment [74]; the transfer of labor and resources out of agriculture provides a fundamental basis for economic growth [75]. As observed for several decades all over the world [76], the agricultural sector in China has changed dramatically since the late 1970s [77]; more and more labor has shifted into highly labor-intensive subsectors of husbandry and fishery in China, and some crops have grown more rapidly by diversifying out of staple grains into higher-valued crops [24] because of the slowed yield growth resulting from land degradation [20] and increasing costs imposed by rising wages and rural labor shortages [78]. Along with the declining share of agricultural employment, a declining GDP from agriculture is inevitable [79] because of higher income demand for non-agricultural goods [80]. China underwent its agriculture modernizing in 1980 mainly by the switch from grain production to higher-value agricultural products as a response to upgrading the food consumption patterns of the Chinese people [81]. As the fastest growing economy in the world, the contribution of the agricultural sector in China has declined from 27.69% in 1978 to 7.11% in 2019, while the contributions from the secondary and tertiary industries have risen to 38.97% and 53.92%. The contribution of the tertiary industry reached 45.46% in 2011; it was the first time that tertiary industry was the biggest contributor to GDP, and the urban population exceeded the rural population for the first time as a result of social transition in China (see Figure 6).

Agriculture consists of farming, animal husbandry, fishery, and forest products. Farm diversification has been proposed to contribute regional agricultural development. Since 1978, there has been a remarkable diversification trend in rural China associated with the emergence of higher-value cash crops and rising livestock production; then, the ratio of the farming sector to the agricultural gross domestic product (GDP) marginally shrunk during the 1980s and then recovered slowly during the period of the 1990s, finally retained steady slow growth. The share of crop subsector to total agricultural GDP has declined from 79.99% in 1978 to 53.29% in 2019, with a sharp drop of 26.7%; the contribution of animal husbandry increased from 14.98% in 1978 to 26.67% in 2019, with the biggest increase of 11.69%; while fishery increased from 2.43% in 1978 to 10.14% in 2019, forestry increased from 1.58% in 1978 to 4.66% in 2019. As extensive diversification has shifted agricultural production away from the traditional subsistence crops to the high-value-crops and modern livestock products [82], the share of livestock and fisheries in total agricultural output value grew from 18.42% in 1978 to 36.81% in 2019, which peaked 44.4% in 2008 (see Figure 7).

Reflected by the rising contributions of animal husbandry and fishery, agriculture in China has undergone major restructuring as a response to technological developments and changing government support policies. Associated with the structural transformation, Chinese food consumption has been shifting from the traditional 8:1:1 ratio of grain: vegetables: meat to a ratio of 4:3:3 [81]. The same occurred in the shift out of cropping into livestock, aquaculture and off-farm employment [82], and more specialized crop–livestock systems [62]. Agricultural productivity in China has experienced rapid growth over the past four decades [83]; its output value per capita has increased from 551.14 in 1978 to 7046.93 in 2019, rising to 1329% of what it had been in 1978; however, as shown in the following figure, there are big gaps among outputs per capita of the three sectors, indicated by 1:7:5 ratio of primary, secondary and tertiary industry productivity in 1978 as compared to the ratio 1:5:4 in 2019 (see Figure 8).

If the price in 1978 was taken as 100, the gross domestic products (GDP) of agriculture has grown steadily from RMB 1578.14 billion in 1978 to 13708.01 in 2019; the aggregate farming products were about 9.81 times larger in 2019 than that in 1978. It is shown that the average annual growth rate fluctuated over time; after negative annual growth rates in 2006, 2009, and 2017, it was followed by a slow recovery with an annual growth rate of 0.3% in 2018 and 6.1% in 2019. The average annual growth rate peaked in 1994 (see Figure 9).

While the accomplishment is impressive, aggregated agricultural productivity growth in China has recently hit a bottleneck [84]. China’s yield of wheat, rice and corn accounts for only 60%, 71% and 67% of the average yield of the top 10 countries with the largest yields, respectively [85]. Global agricultural production has been massively increased by a huge specialization and simplification of agricultural systems [86], which led to a quantum jump in the share of the animal husbandry subsector during the 1980s. Later, although the value of livestock (at constant prices) nearly double during the 1990s; its share in agriculture remained stagnant at 20.54% in 1989, and the fisheries subsector has increased by about 50% during 1990s. However, the real contribution of four subsectors of agriculture did not change much during the same time period. The contribution from crop subsector decreased from 79.99% in 1978 to 75.621% in 2019; at the same time, the shares of forestry, animal husbandry and fishery increased from 3.44% to 4.41%, 14.98% to 16.54%, 1.58% to 3.43% (see Figure 10).

Although increasing uncertainties about the climate and negative impacts arising from agricultural intensification have put into question the ‘‘specialization—higher productivity’’ path of development, the same tendency has emerged in agroecosystems in less developed regions to achieve market competitiveness of specialization at the farm level. It is believed that gains from specialization and regional comparative advantage have not been fully achieved in China [87]; the regional patterns of China’s grain production are far from representing regional comparative advantage [88,89]—this is partly proved by our study. It was shown that, taking into consideration of the impact of price, the contribution of four subsectors has not changed much (see Figure 10).

Diversification can be broadly divided into two stages [90,91]. Initial diversification is at the cropping level, with a shift away from monoculture to mixed crops. At the second stage, diversification is understood as being mixed farming [92] with a shift of resources from one crop (or livestock) to a larger mix of crops (or livestock) or a mix of crops and livestock [93]. What are the patterns of AD across regions in China? The following analysis of the dynamic character of AD across regions will highlight insightful views on future agricultural transformation.

## 3. Stages of Agricultural Diversification across Regions

### 3.1. Stages of Regional Agricultural Diversification

Taking into consideration of the fact that data on agricultural inputs, crop production and the agricultural economy are officially recorded according to administrative regions; they are different from the agricultural zones, which are classified based on their natural conditions, crop suitability and levels of agricultural productivity [12]. This section uses data from the main administrative regions to illustrate the diversity of agricultural production across China.

The regions are divided as follows: the northeast consists of Heilongjiang, Jilin and Liaoning provinces; the north central region covers Hebei, Shanxi, Inner Mongolia and the cities of Beijing and Tianjin; the eastern region covers Shanghai, Jiangsu, Zhejiang, Anhui, Fujian, Jiangxi and Shandong; the central south region consists of Henan, Hubei and Hunan, the south region consists of Guangdong, Guangxi and Hainan; the southwest covers Chongqing, Sichuan, Guizhou, Yunnan and Tibet; the northwest covers Shaanxi, Gansu, Qinghai, Ningxia and Xinjiang.

The regional AD index is measured as follows (Equations (13) and (14)):(13)$$D_{it}=1-H_{it}=1-\sum_{n}(Q_{nit}/{\sum_{k}Q}_{kit}{)}^{2}$$
(14)$$p_{it}=\sum_{n}(Q_{nit}/\sum_{k}Q_{kit}{)}^{2}$$
where *Q_nit_* is output value of subsector *i* in district n in year *t*. District includes northeast, northcentral, eastern, central south, south, southeast and northwest.

A smaller *D_it_* implies that agriculture GDP of the district is more concentrated. We can see that eastern region has the highest AD level and highest per capita GDP. The south region reached the highest turning point as early as 2003; it was followed by the southeast region, the northeast, and the northwest. By contrast, the central south region reached the highest AD level in 2017, and the northcentral and eastern regions reached the highest AD level in 2019. Figure 11 shows that in 1978, the sets of agricultural diversification indices and per capita GDP (RMB/person) across regions were as follows: south (0.46, 290), southeast (0.38, 238), north central (0.35, 478), northwest (0.34, 322), eastern (0.34, 403), central south (0.31, 274), northeast (0.30, 506). In 2019, the sets of agricultural diversification indices and per capita GDP across regions were as follows: eastern (0.64, 90,643), south (0.62, 77,021), northcentral (0.60, 67,599), northeast (0.57, 46,552), central south (0.57, 62,192), southeast (0.54, 55,045), and northwest (0.43, 52,974). Except for the northeast, where economy suffered sudden slow down recently, the AD indices in other regions increased in association with rising per capita GDP.

The two selected regions with the highest and lowest AD, the eastern and northwest regions, are used to further examine the stages of AD across China. The eastern region was highly diversified over time; its AD levels ranged from 0.34 in 1978 to 0.64 in 2019, the highest turning point of 0.64 with per capita GDP of RMB 90643 /person in 2019 (see Figure 11). With the highest per capita GDP, the contribution of the crop subsector in Shanghai decreased from 82.93% in 1978 to 50.81% in 2019, while animal husbandry increased from 11.93% in 1978 to 24.94% in 2019. Jiangsu had a modest diversification level and its per capita GDP ranked second in this region. The contribution of the crop subsector in Fujian decreased from 77.68% in 1978 to 38.28% in 2019, while animal husbandry increased from 10.51% in 1978 to 19.75% in 1994, and to 19.72% in 2019. By contrast, for Shandong province, the contribution of crop subsector decreased from 82.93% in 1978 to 50.81% in 2019, while animal husbandry increased from 11.93% in 1978 to 32.7% in 1994, and to 24.94% in 2019 (see Figure 12). The development of animal husbandry plays an important role in shifting agricultural diversification into the third stage of mixed farming and cropping systems in the eastern region.

The AD level of the northwest ranged from 0.34 in 1978 to 0.4303 in 2019; the turning point of 0.47 was reached in 2008 with per capita GDP of RMB 17422 /person (see Figure 11). Gansu, Qinghai and Xinjiang had reached the turning points of AD in 1988 to 1989. Shaanxi and Ningxia had reached their turning points of AD in 2003. In the northwest, the two provinces with lower AD indices of 0.39, Gansu and Xinjiang, deserve special attention. In 1988, Gansu reached its highest diversification level of 0.47 when its per capita GDP was RMB 898.13/person, and reached lowest AD (0.39) in the whole of China in 2019, where the farming sector was the main contributor of agricultural products. Farming’s agricultural contribution ranged from 80.4% in 1978 to 69.21% in 2019; forestry contributed 2.76% in 1978 to 2.02% in 2019; animal husbandry’s contribution ranged from 16.88% in 1978 to 20.96% in 2019; fishery ranged from 0.02% in 1985 to 0.11% in 2019. Xinjiang peaked its diversification index of 0.44 in 1989 when its per capita GDP was RMB 1494.43/person; its diversification level in 2019 was 0.41. In Xinjiang, farming’s agricultural contribution ranged from 74.53% in 1978 to 67.94% in 2019; forestry contributed 1.78% in 1978 to 1.7% in 2019; animal husbandry’s contribution ranged from 23.57% in 1978 to 23.77% in 2019; fishery ranged from 0.16% in 1978 to 0.71% in 2019(see Figure 13). In the northwest region, the turning points of different provinces had come in the late 1990s when farming subsector was the main contributor of agricultural products with a relatively lower diversification index.

Generally speaking, the AD levels varied across regions associated with different performance of economic growth. Although horticulture and livestock products can provide more employment opportunities [94] and more benefits [95] than the traditional crops, agriculture diversification requires further support of technological advancement, better infrastructure, and well-functioning agricultural markets [9]. The extent of AD is still increasing, even with the reemphasis on modernizing agriculture in the 2017 No.1 document. Since 1988, the central government has implemented mayoral responsibility for vegetable baskets in order to ensure people’s livelihoods, stabilize prices, and supply vegetables and meat. Since 1994, for the same purpose, the central government has implemented a system in which governors are responsible for the price of rice bags. All these partly explain the rising AD in some regions. The next section will examine the geographic distribution of agricultural subsectors across regions, which is the basis for deeply understanding the relationship between the patterns of AD and agricultural modernization.

### 3.2. Geographic Diversification of Agricultural Subsectors in China

Another diversification index can be calculated as (Equations (15) and (16)):(15)$$D_{nt}=1-H_{nt}=1-\sum_{i}(Q_{nit}/\sum_{j}Q_{njt}{)}^{2}$$
(16)$$p_{it}=\sum_{i}(Q_{nit}/\sum_{j}Q_{njt}{)}^{2}$$

*D_nt_* is the measure of diversification for a particular district in year *t*. The same as in equation (10), *Q_nit_* is output value of subsector *i* in district n in year t. The subsectors of agriculture consist of crops, livestock, fisheries and forestry. *D_nt_* is plotted in Figure 8 for the four subsectors of broad agriculture in China. According to the value of *D_nt_*, the index of the crop subsector ranged from 0.94 in 1978 to 0.95 in 2019; the index of animal husbandry was from 0.94 in 1978 to 0.95 in 2019; the index of forestry rose from 0.94 in 1978 to 0.96 in 2019; the index of fishery went through up and down shocks, ranging from 0.88 in 1978 to 0.92 in 2019, increasing by 0.04. Fishery is with the relatively least dispersed regional distribution, while the other three sectors have nearly same level of dispersed regional distribution. The most dynamic change is seen in the fishery sector, while the output value of fisheries was the most concentrated among four sectors (see Figure 14).

#### 3.2.1. Geographic Distribution of Forestry GDP across Regions

It was shown that there were three nuclei of forestry in Sichuan, Hunan, and Fujian. Forestry GDP was mainly concentrated in central and southern parts of China, but since 1978, the center of GDP moved gradually from the southeast to the central west of Sichuan, Hunan and Fujian, and their forestry GDP proportions rose year by year; the three regions’ forestry GDPs in 1978 were 6.65%, 6.49% and 4.80%, respectively; the proportions reached 8.40%,7.46% and 7.23% in 2019. By contrast, the proportions of forestry in Jiangxi and Guangdong gradually decreased from 12.20% and 10.35% in 1978, to 5.94% and 7.07% in 2019, respectively.

The GDP of forestry indicated a trend of regional diversification, and the center steadily moved to the southeast of China’s territory (See Figure 15). The contribution of forestry to total agricultural products increased over time associated with increase in the area of economic forest. After the enhancement of afforestation policy, the afforestation area peaked in 2003 with implementation of the Conversion of Cropland to Forest Program (CCFP) in 2002, which was designed to establish forests and grasslands on all vulnerable sloping lands in China, and a special focus was given to the Yellow River and Yangtze River regions [96,97]. As most planted trees there were ecological forest, the forestry GDP in Shaanxi declined from 2.43% in 1978 to 1.84% in 2019.

#### 3.2.2. Patterns of Fishery Concentration across Regions

The Chinese government’s priority for food security has led China to commit considerable resources to agricultural and fish research on feeds, fish species, and farming practices. The marine fish production improved due to the rising production of inland fisheries over time because of a well spread location of rivers, canals and reservoirs. Compared with the other three subsectors, the concentration of fishery, which was located in the southeastern coastal areas of China, mainly in Jiangsu, Guangdong, Shandong, Fujian, and Zhejiang, was relatively stable; their contributions in 2019 were 13.85%, 12.13%, 11.11%, 10.83%, and 8.60%, respectively. In the past 40 years, the most obvious change was that of the status of fisheries in Hubei Province; its proportion of the fisheries in the whole country increased from 2.63% in 1978 to 5.03% in 1998, and then further increased to 9.17% in 2019 (See Figure 16).

According to the China Fishery Statistical Yearbook, marine fisheries could be divided into mariculture, marine fishing and distant water fishery. In 1978, mariculture in China was primarily located in the Provinces of Guangdong (24.52%), Shandong (15.61%), Zhejiang (15.75%), Jiangsu (13.85%), Liaoning (9.50%), and Fujian (8.96%). Mariculture had spread to all coastal areas. Besides the traditional areas of Jiangsu (13.85%), Guangdong (12.13%), Shandong (11.11$), Fujian (10.83%) and Zhejiang (8.6%), Guangxi and Anhui also experienced rapid growth of fishery products in China.

The fisheries subsector has diversified over time as a response to the gradual shift from marine to inland fisheries, both in fresh and brackish water. The regional distribution of mariculture has shifted from integrated development in traditional areas to common development in all coastal areas. Fisheries in China have experienced a shift from fishing-dominated to aquaculture-dominated sectors [98]. Based on statistical data from various years, the share of seawater production in the total aquatic products fell from 77.25% in 1978 to 50.65% in 2019, while that of inland fisheries rose from 22.75% in 1978 to 49.35% in 2019. A bulk of the growth in culture fisheries has come from the fresh water aquaculture; the share increased from 26.04% in 1978 to 78.38% in 2019. The dispersed expansion of inland fisheries has exerted concerns over negative impacts on the salinity of coastal areas.

#### 3.2.3. Patterns of Farming Agricultural Concentration across Regions

During these four decades, the GDP of the agricultural sector has been concentrated in central China and at both ends of China’s territories. Agricultural GDP was mainly concentrated in the central region since reform and opening up; Sichuan (including Chongqing), Henan, Shandong have always been the highly productive nuclei of farming GDP, remaining in the top three. In 1978, Jilin, Shandong, and Henan’s agricultural GDP were 7.62% 7.59% and 7.31%, respectively; in 1998, they accounted for 2.77%, 8.57%, and 8.14%, respectively. In 2019, their contributions were 5.80%, 7.44%, and 8.19%, respectively. In 2019, Sichuan province ranked the first place by replacing Jilin. As time went by, the proportions of the crop subsector in Xinjiang and Heilongjiang kept rising; by contrast, the proportion of agricultural GDP in Hebei, Jiangsu, Hunan, Hubei and Guangdong declined. The traditional agricultural nuclei was Jilin, with 7.62% to total agricultural GDP in 1978; it was 5.80% in 2019 (See Figure 17).

#### 3.2.4. Patterns of Animal Husbandry Concentration across Regions

Last but not least, growing at a fast rate, the share of the livestock sector in the total value of agricultural output is progressively rising in China. Figure 18 indicates a trend of the gradual dispersion of animal husbandry, and the difference in the colors of the 30 regions gradually decreases, but Sichuan has always been the main contributor of animal husbandry GDP in China. The proportion of Sichuan’s animal husbandry to GDP in the country was 8.71% in 1978, 9.15% in 1998, 9.64% in 2018, and 10.06% in 2019. Except Beijing and Tibet, all provincial contributions of animal husbandry increased over time. Meanwhile, another two top provinces were Shandong and Henan; their shares to animal husbandry GDP were 5.82% and 5.19 in 1978, and 7.30% and 7.01% in 2019, respectively. The higher AD of animal husbandry is mainly due to a large share of poultry and hogs in the total value of livestock products.

Historically, small-scale operations typified Chinese agriculture. Although it is still dominated by small operations, hog production in China has become more concentrated in large commercialized operations, especially in the periurban areas of big cities [99]. The situation of concentration in periurban areas of big cities together with structural changes pose a critical threat to environmental management. In China, perhaps relocation is an adaptive response to regulatory technical standards prescribed by the government, instead of a response to changing market circumstance. With their main focus on producing animals, large commercial facilities intend to purchase most of their feed from off the farm, and the concentration of industrialized livestock manure on a small, local landmass will lead to runoff of phosphorus, nitrogen, and other pollutants [100,101]. The situation of being confined in the periurban areas of big cities, together with structural changes, pose a critical threat to environmental management; the coordinative development of the farming agriculture and animal husbandry sectors are of particular importance for China in the pursuit of agricultural modernization.

## 4. Conclusions

AD in China is gradually shifting to the agrosystem of livestock/fishery activities. In the beginning, the degree of agricultural diversification in the southwest was relatively high, and then the center of agricultural diversification gradually moved to the southeast; after that, in 2019, the degree of agricultural diversification in the economically developed eastern provinces was obviously higher than those in other regions. As AD levels rose over time, the U-shaped relationship between AD and per capita GDP was not always confirmed because of the dynamic stages of AD. In most regions, the higher the per capita GDP, the higher the AD. Various provinces had different patterns of AD with varying turning points of the diversification curves along with improvement of the economic situation. It was seen that some provinces in the eastern and central south regions moved toward either increasing diversification in one direction, and other provinces changed direction, first moving toward diversification and later toward concentration or vice versa. The oscillations of AD among the provinces through time mean a cyclic tendency from the point of view of diversification across regions.

The patterns of diversification differ across regions due to diversified agricultural subsectors resulting from different natural and socio-economic circumstances. Regionally, the patterns were shifting from crop-dominance to the livestock subsector during the 1980s and 1990s. The exceptions were the eastern and northeastern regions, where the ratios of both crop and livestock subsectors to total production of agricultural outputs rose while the contribution of fisheries and forestry declined. In the southern region, the share of fisheries and forestry in total output was declining during the 1980s and 1990s. The crop subsector was the main source of rural income in agriculture followed by the livestock sector, especially in less developed regions with lower agricultural diversification levels, where the crop subsector persistently dominated the leading position. Due to the strong complementary synergy of crop and livestock production, it is imperative to facilitate the coordinative development of the two subsectors of crop subsector and animal husbandry during the process of agricultural transformation.

As a populous country with limited arable land, it is a challenging issue for Chinese agriculture to continue to increase food production and protect degraded environments. To reconcile the need of food production with the requirement for environmental conservation, ecological intensification of agriculture is a feasible option for small farms to provide both high socio-economic outputs and multiple environmental benefits. However, the successful ecological intensification of agriculture does not mean diversification at all levels of the organization of the farm and the region [86]. In China, the negative environmental impacts of modern agriculture constrained in small scale farms are due more to an excess use of chemical and fertilizer inputs instead of specialization and uniformity in pursuing an excess of productivity. It is urgent for Chinse government to reconcile the increase of agricultural productivity via specialization and scale economy at both farm and regional levels with the environmental conservation through ecological intensification of agriculture.

Based on mesolevel data, our study has examined the patterns and dynamic character of AD at the regional and national levels. Further studies are needed from both macro and micro perspectives. Firstly, from macro perspective, it is argued that there exists an inverted U-pattern relationship between diversification and economic growth in the aggregate economy and the manufacturing sector; however, this is only partly proved by our study on the agricultural sector in China. The AD index remains relatively steady for over more than 17 years after the turning point, and the causes underlying the steady trend need further study.

Secondly, from the micro perspective, the determinants of famers’ diversification strategies and their impacts on farmers’ welfare and on the ecosystem have been ignored in our study due to the lack of microdata at the household level. In fact, different production units have different strategy options [102]; it is the farmers who decide whether to diversify or not [103], and AD does have an impact on farmers’ welfare and adoption of technology [104] and on the resilience of the agrosystem [105]. Therefore, factors underlying farmers’ decisions for agricultural diversification, especially the impact of government policy on farmers’ options [106], deserve special attention, and the potential impacts of regional and national agricultural diversification on ecosystem functions require a thorough study in the future.

## Figures and Tables

**Figure 1 ijerph-18-04978-f001:**
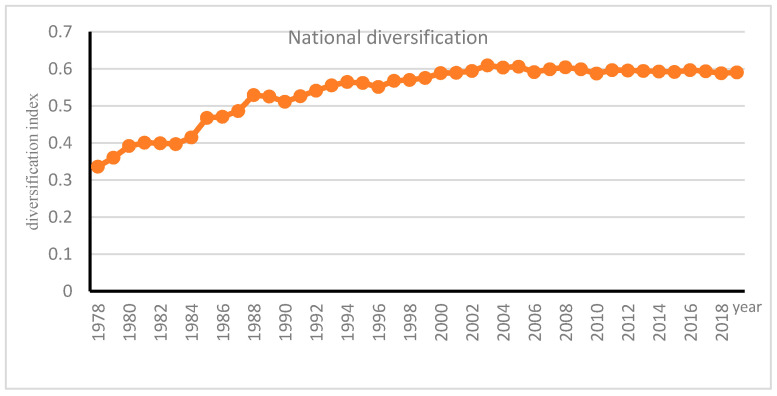
Trend of AD in China.

**Figure 2 ijerph-18-04978-f002:**
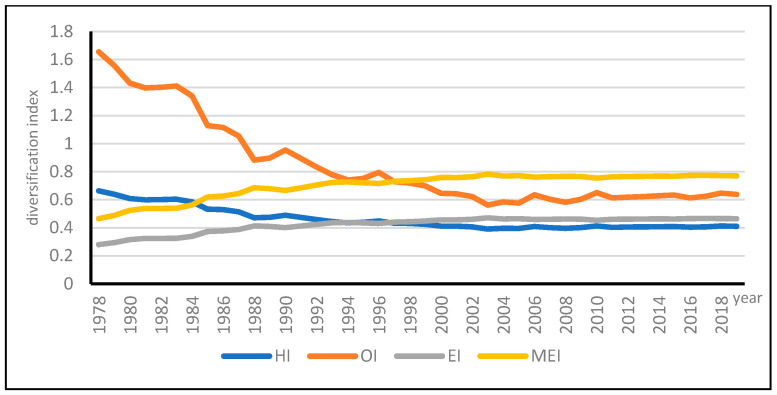
Different measurements of agriculture diversification.

**Figure 3 ijerph-18-04978-f003:**
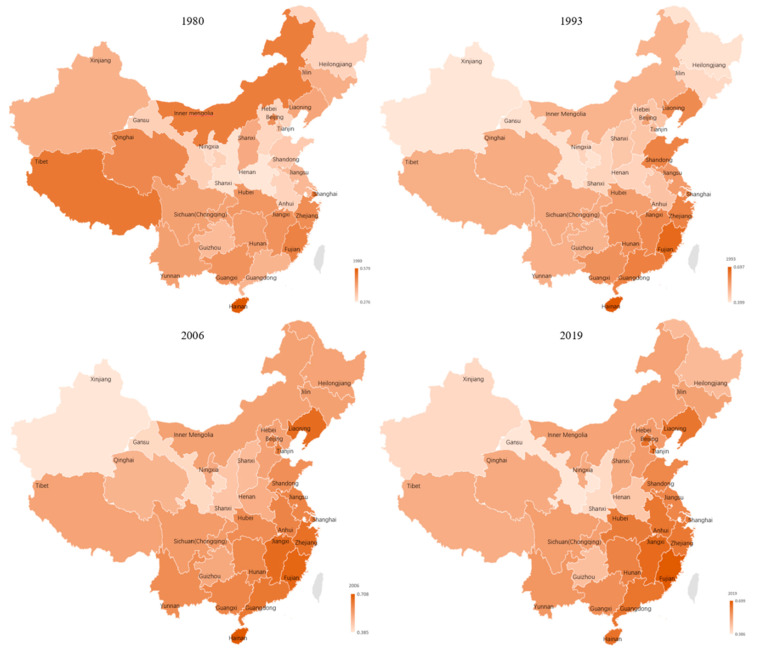
Geographical distribution of AD in China (1980–2019).

**Figure 4 ijerph-18-04978-f004:**
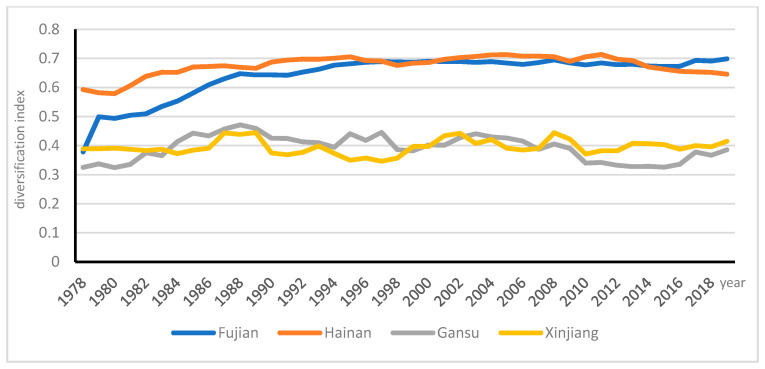
Trends of selected four provinces of highest and lowest diversification indices.

**Figure 5 ijerph-18-04978-f005:**
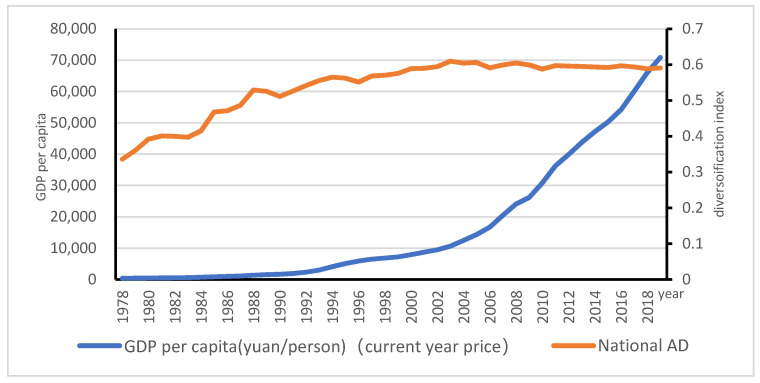
The relationship between AD and economic growth.

**Figure 6 ijerph-18-04978-f006:**
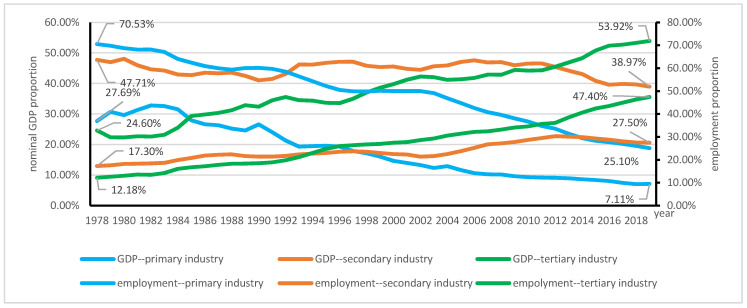
The composition of nominal GDP and employment of three industries in China (1978–2019).

**Figure 7 ijerph-18-04978-f007:**
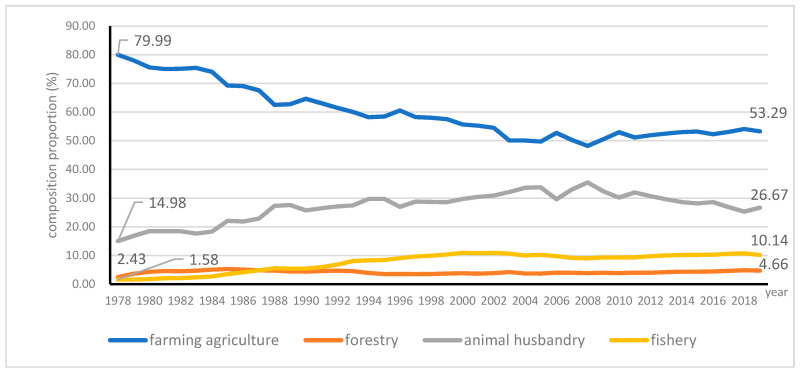
The composition of agricultural sectors.

**Figure 8 ijerph-18-04978-f008:**
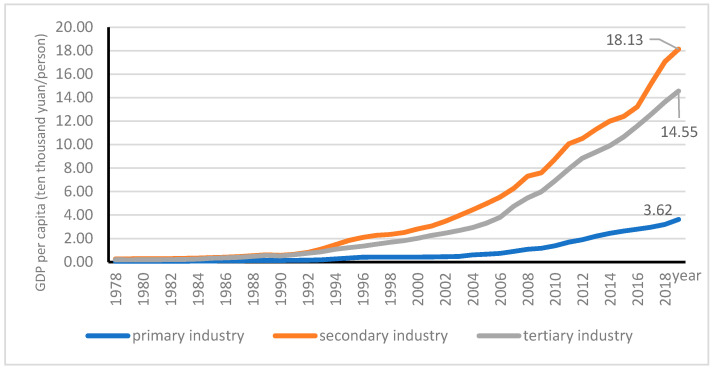
Annual per capita GDP output of three industries.

**Figure 9 ijerph-18-04978-f009:**
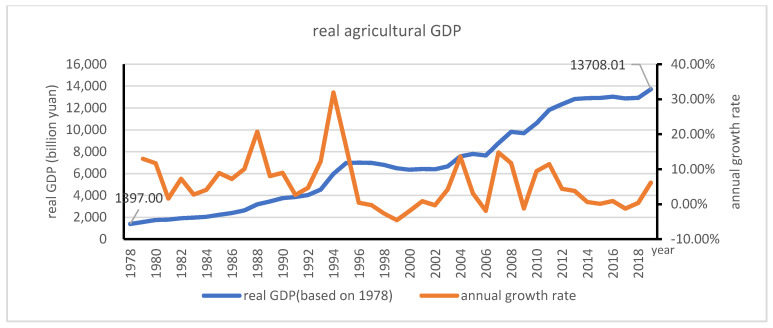
Development of broad agriculture in China.

**Figure 10 ijerph-18-04978-f010:**
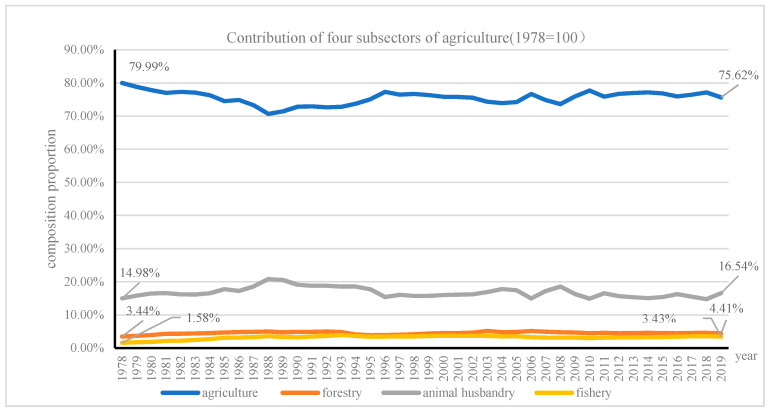
Contribution of four subsectors of agriculture measured at price of 1978.

**Figure 11 ijerph-18-04978-f011:**
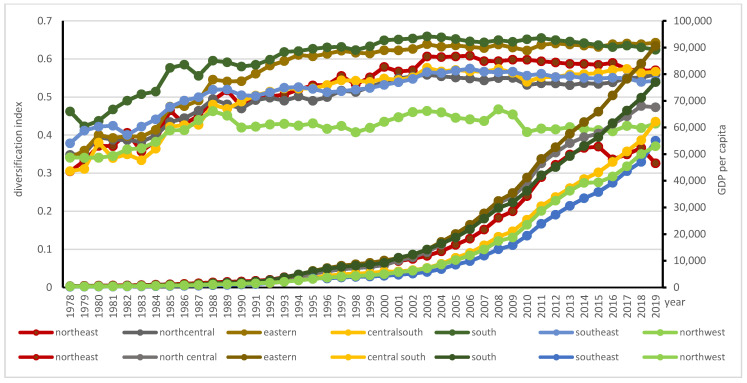
Geographic distribution of AD across regions in China.

**Figure 12 ijerph-18-04978-f012:**
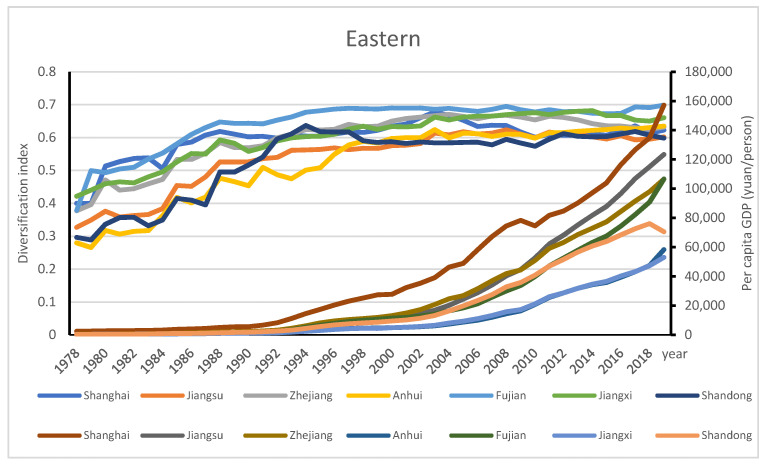
Geographic distribution of AD in the eastern region of China.

**Figure 13 ijerph-18-04978-f013:**
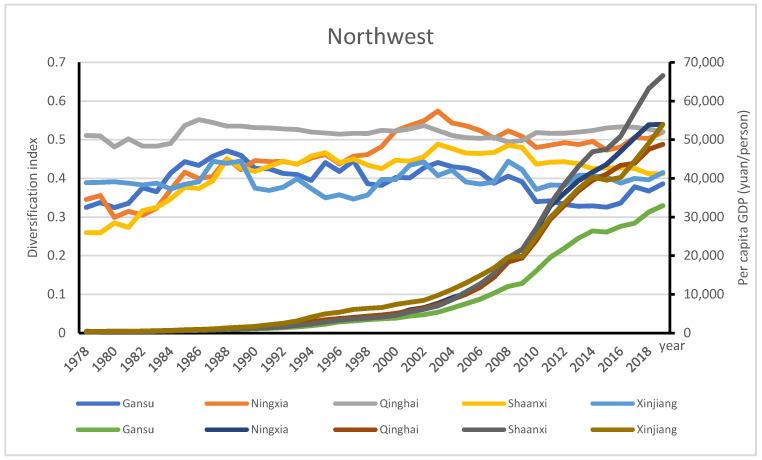
Geographic distribution of AD in the northwest of China.

**Figure 14 ijerph-18-04978-f014:**
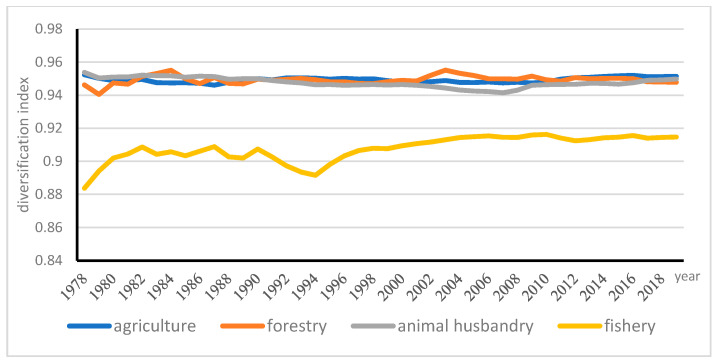
Diversification indices of four subsectors in China.

**Figure 15 ijerph-18-04978-f015:**
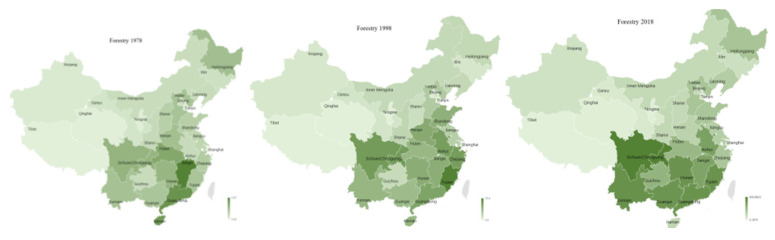
Geographic distribution of forestry GDP in 1978–2018.

**Figure 16 ijerph-18-04978-f016:**
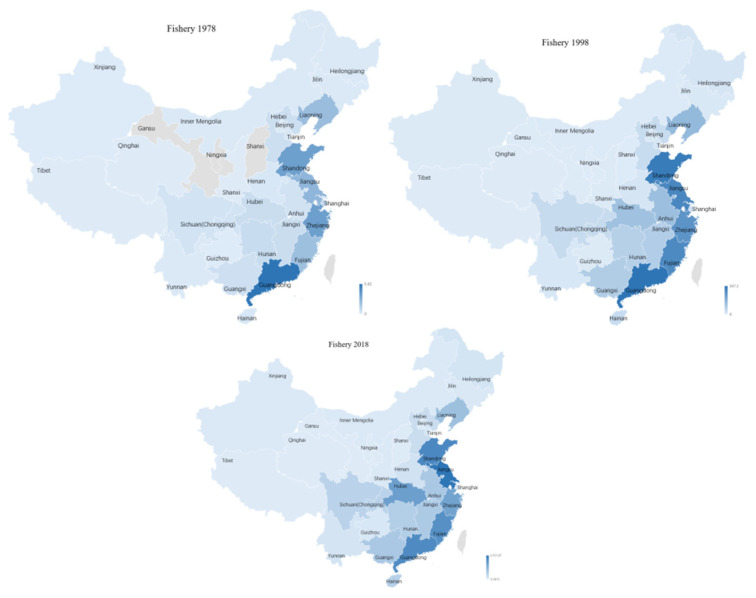
Geographic distribution of fishery GDP in 2018.

**Figure 17 ijerph-18-04978-f017:**
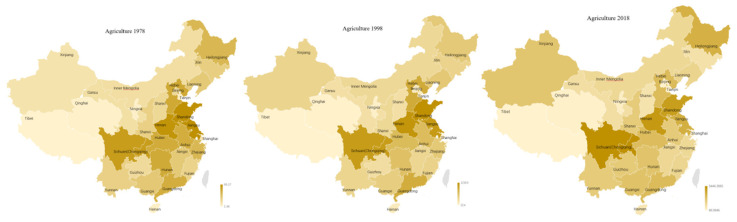
Geographic distribution of farming subsector GDP in 1978–2019.

**Figure 18 ijerph-18-04978-f018:**
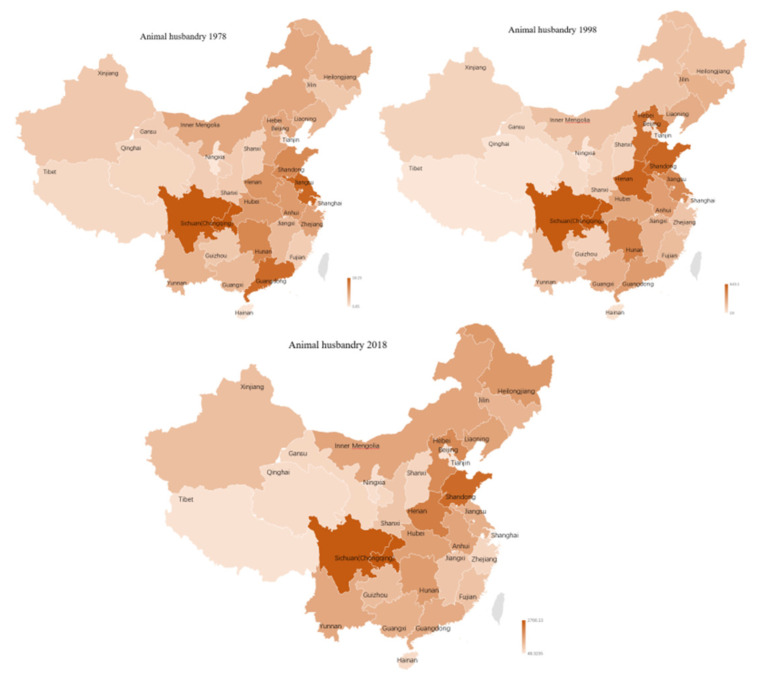
Geographic distribution of animal husbandry GDP in 1978–2018.

## Data Availability

Not applicable.

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
