# Peer review of "Patterns of Agricultural Diversification in China and Its Policy Implications for Agricultural Modernization"

_ijerph, 2021, doi:10.3390/ijerph18094978_

Round 1
Reviewer 1 Report
Overall, this is a comprehensive study for which the author(s) deserve(s) a big thank you. However, there are several areas that need improvement: (i) Some references are very old of 60s and 70s. Even though those were the time such concepts were endangered, there could be other studies undertaken based on those theories and those succeeding studies could have additional insights. If examples from other countries, if available, could be included in the background or even in the discussion, that would be very insightful to the readers. (ii) I would suggest the author(s) to provide a few examples (beyond equations) for how different indices (e.g., HI) were calculated. (iii) Some sentences or write up should be rewritten to make sure the contents flow well. (iv) Language editing is required at seveal places. (v) There are several inconsistencies in the referencing. Make sure they follow the journal's referencing style, and (vi) finally, I would suggest the author(s) to include messages for other countries based on this study.
Author Response
Response to Reviewer 1 Comments
Thank you for your helpful comments.
Point 1: Some references are very old of 60s and 70s. Even though those were the time such concepts were endangered, there could be other studies undertaken based on those theories and those succeeding studies could have additional insights. If examples from other countries, if available, could be included in the background or even in the discussion, that would be very insightful to the readers.
Response 1: As you pointed out that some references are too old of 60s and 70s.Therefore, we added some new references in recent years in the last paragraph. Here I just want to say that some successful examples of Asian Countries of Japan and Thailand, and European countries have been provided in the background information. We also added some new literature in last paragraph of the paper, which outlined shortages of this study and pointed out possible areas for future study.
Point 2: I would suggest the author(s) to provide a few examples (beyond equations) for how different indices (e.g., HI) were calculated.
Response 2: Following is an example for the calculation of the district diversification index under subsector , it is given as follows:
, (7)
Whereis output value of subsector in district n in year t, they are farming, forestry, animal husbandry, and fishery.
If further information is required, we are happy to provide all data.
Point 3: Some sentences or write up should be rewritten to make sure the contents flow well.
Response 3: Yes, it is helpful to rewrite some sentences, it was included in the new version of the paper. Special attention was paid to the abstract and conclusions.
Point 4: Language editing is required at several places.
Response 4: We have read the paper thoroughly to correct misspelled words. Sorry for the troubles caused by carelessness of authors. There are some misspellings of words, such as agriculture, it is not forgivable.
Point 5: There are several inconsistencies in the referencing. Make sure they follow the journal's referencing style.
Response 5: We have updated all references.
Point 6: finally, I would suggest the author(s) to include messages for other countries based on this study.
Response 6: Messages for other countries were included in the part of introduction, they are Asian and European countries. We also added some new literature in last paragraph of the paper, which outlined shortages of this study and pointed out possible areas for future study.
Reviewer 2 Report
Dear Authors,
The study aims at investigating the "Patterns of agricultural diversification in China and its policy implications for agricultural modernization".
Although the paper deals with an interesting topic, major revisions are required.
The paper is not satisfactory written, needs a careful editing, fonts, and style.
Further, the study aim and background are not well presented, repetitions occurring in the paper should be avoided.
However, it is recommended:
- Reformulate the abstract by telling prospective readers what you did and what the important findings of your research were.
- Introduction can be improved in order to show better aim.
- Please carefully consider and revise the logic of some parts.
- Carefully check the full text.
- Please do more to highlight how the work advances or increments the field from the present state of knowledge and provide a clear justification for your work.
- Methodology be supported with reference literature.
- Conclusion section needs improvement. Please provide more quantitative key contributions of the study with proper discussions, highlight the limitations of this study and the future work.
- English proofreading is needed. Some description is not professional for a scientific article.
Accordingly, it is opinion of this reviewer to accept with major revisions the proposed manuscript for a publication on this journal.
Author Response
Response to Reviewer 2 Comments
Thank you for your valuable comments.
Point 1: The paper is not satisfactory written, needs a careful editing, fonts, and style.
Response 1: We have read the paper thoroughly in order to correct misspelled words. We have corrected all misspelled words.
Point 2: Further, the study aim and background are not well presented, repetitions occurring in the paper should be avoided.
Response 2: We have rewritten the study aim and background to make it concise, repetitions occurring in the paper have been corrected.
Point 3: Reformulate the abstract by telling prospective readers what you did and what the important findings of your research were.
Response 3: Yes. We have rewritten the abstract by using more connective words, which makes it easy to be read.
Point 4: Introduction can be improved in order to show better aim.
Response 4: We have rewritten the study aim and background to make it concisely.
Point 5: Please carefully consider and revise the logic of some parts.
Response 5: Yes, it is a helpful suggestion. We have updated all parts of the paper in the tracked new version.
Point 6: Carefully check the full text.
Response 6: We have checked the full text to delete all parts of possible misspelling and repetitions.
Point 7: Please do more to highlight how the work advances or increments the field from the present state of knowledge and provide a clear justification for your work.
Response 7: Yes. As mentioned in the manuscript, “Studies concerning AD at the meso-level are limited [3]. This study aims to fill this this space by empirically examining the extent of AD regional levels and national level. Facing the dilemma of rural development and environmental conservation, China is a good study area as it is a large and diverse country”.
Point 8: Methodology be supported with reference literature.
Response 8: To provide necessary support, we have reviewed some articles on the measurement of diversification, including the literature oldest and new in this area.
Point 9: Conclusion section needs improvement. Please provide more quantitative key contributions of the study with proper discussions, highlight the limitations of this study and the future work.
Response 9: Yes, we have changed the title of last section. In the last paragraph, we highlight the limitations and the future work.
“Based on meso-level data, our study has examined the patterns and dynamic character of AD at regional and national levels. However, determinants on famers’ diversification strategy and its impacts on farmers’ welfare and ecosystem have been ignored in our study due to the lack of micro data at household level. In fact, different production units have different strategy options in terms [102], it is farmers who decide whether to diversify or not [103], AD do have impacts on farmers’ welfare and technological adoption [104] and on the resilience of agrosystem [105]. Factors underlying farmers’ decision of agricultural diversification, especially the impact of government policy on farmers’ options [106], deserve special attention, and the potential impacts of regional and national agricultural diversification on ecosystem functions require a thorough study in the future.”
Point 10: English proofreading is needed. Some description is not professional for a scientific article.
Response 10: We have checked the full text to delete all parts of possible misspelling and repetitions.
Reviewer 3 Report
This study seeks to analyze the facet of Chinese agricultural revolution through an examination of agricultural diversification at both national and provincial levels. The authors discuss different measures of agricultural diversification and apply one of them to analyze agricultural diversification and its evolution in China from 1978 to 2019.
This paper addresses a topic of great interest for the sustainability of agricultural activity and for society in general.
I think the paper is well written. However, I understand that this paper is very descriptive and that it should contribute something else:
- In view of the description presented on the geographical distribution of the agricultural activity, I understand that a contribution of the work could be related to the explanatory factors of this distribution and its evolution. A regression model could be proposed to identify and assess such factors, analyzing possible spatial effects.
- I believe that the inverse u-shaped relationship between agricultural diversification and GDP per capita should be analyzed through a regression, as other studies do (Kalemli-Ozcan et al., 2003), also considering the appropriate control variables.
- The discussion of the results and their implications for political decision-making and farm management should be deepened. The discussion and conclusions must adequately justify the contributions of the paper based on the previous literature.
On the other hand, you must take care of the writing of the work and the formal aspects. For example:
- At the end of line 34, correct the word agricultural.
- At the end of line 81 the word "this" is duplicated
- Review the formulas in section 2.1. In the first formula is pj and I think it would be pi. In the second figure say and in the next line dj ...
- The y-axis title should appear in the figures.
Author Response
Response to Reviewer 3 Comments
Thank you for your helpful comments.
Point 1: In view of the description presented on the geographical distribution of the agricultural activity, I understand that a contribution of the work could be related to the explanatory factors of this distribution and its evolution. A regression model could be proposed to identify and assess such factors, analyzing possible spatial effects.
Response 1: This paper aims to reveal the dynamic evolution of agricultural diversification and outline the patterns of agricultural diversification. Therefore, the dynamic character of agricultural diversification is the main focus of this paper. Anyway, your valuable suggestion is highly appreciated, a regression model will be included in next paper due to the worry about space limitation in one paper.
Point 2: I believe that the inverse u-shaped relationship between agricultural diversification and GDP per capita should be analyzed through a regression, as other studies do (Kalemli-Ozcan et al., 2003), also considering the appropriate control variables.
Response 2: It is a good suggestion to examine empirically the relationship between agricultural diversification and GDP per capita, also considering the appropriate control variables. But here, this paper aims to reveal the dynamic character of agricultural diversification with a descriptive figure, which provides meaningful information. Of course, it is a good research point for future study.
Point 3: The discussion of the results and their implications for political decision-making and farm management should be deepened. The discussion and conclusions must adequately justify the contributions of the paper based on the previous literature.
Response 3: We have rewritten the conclusions.
Point 4: On the other hand, you must take care of the writing of the work and the formal aspects. For example:
- “At the end of line 34, correct the word agricultural”.
Response 4: We have corrected all misspelled words.
Point 5: “At the end of line 81 the word "this" is duplicated”,
Response 5: we have deleted it.
Point 6: Review the formulas in section 2.1. In the first formula is pj and I think it would be pi.
(1)
Response 6: wheredenotes the ratio of j th business in a firm, is the weight, and is named as the number of the businesses in the firm [34,35].
Point 7: In the second figure say and in the next line dj ...
Response 7: Hereis not a satisfactory index because it accounts only for the share distribution between primary products and the aggregate of all secondary products, but quiet with the number of products [37].
Point 8: The y-axis title should appear in the figures.
Response 8: The y-axis titles have been presented in this version.
Reviewer 4 Report
Please consider changing the title of the last section "Brief conclusions and its implications for agricultural transformation" to "Conclusions". Especially since this chapter is not particularly short.
Please rephrase part of the Conclusions chapter to emphasize the importance of the analyzes carried out in the manuscript.
Please adapt the manuscript to the journal's requirements, e.g.:
- unnumbered mathematical formulas
- no spaces before brackets "specialization [4]" - similar errors are found in many places
- page number provided unnecessarily - line 75 "p.249"
In References, there are too few references to the latest scientific publications (from 2019, 2020 and 2021). Only found one from 2020.
Please adapt the References to the journal's requirements (italics, bolding of the year of publication, unnecessary spaces before commas, etc.)
Please consider whether it is necessary to refer to the same source more than once with the same paragraph:
- "In 2007, agricultural modernization was first put forwardby Document No.1in China, which is typically characterized as “specialization, intensification andscale enlargement” [6]as afeasibleoption for smallholder householdsto increase income[5,14]. However, increasedspecialization and broader scalescan alsoendanger ecosystem functions at broader spatial scales [5]; furthermore, agricultural specializationis constrained byinfeasible technologies[15, 16-17]. In fact, non-specialized production is still the main modein China’s agriculture, which inevitably impedes the process of modernizing agriculture [18]. It is imperative for policy-makersto balance their focus on efficientproductionof agricultural modernization, the maintananceof ecosystems,and prosperous rural areas pursued by AD [6]."
- "As the world’s most populous country,tackling food security in China has always been her policy priority [19], which has led to environmental deteriorationarising fromoveruse of chemical inputs [20,21-22]. What are the trends ofAD in the pursuit of agricultural modernizationin China?The existing studieshave mainly focus onthe role of diversification in technology, market rationale and risk reducing strategy [23]. To the author’s knowledge, no empirical measurement of diversification has been applied tothestudy of China’s structural change and development[9], which is a basis for policy formation in restructuring China’s economy to address social inequality and protect the environment[22]."
Author Response
Response to Reviewer 4 Comments
Thank you for your helpful comments.
Point 1: Please consider changing the title of the last section "Brief conclusions and its implications for agricultural transformation" to "Conclusions". Especially since this chapter is not particularly short.
Please rephrase part of the Conclusions chapter to emphasize the importance of the analyzes carried out in the manuscript.
Response 1: We have changed the titel of the last section as conclusions.
Point 2: unnumbered mathematical formulas
Response 2: We have numbered all mathematical formulas in the paper.
Point 3: no spaces before brackets "specialization [4]" - similar errors are found in many places
Response 3: We have added spaces for brackets.
Point 4: page number provided unnecessarily - line 75 "p.249"
Response 4: Page number of p.249 is kept because of the quotation of the literature. It is believed that China is challenging the earth’s environment [25] and deeply affect “global patterns of resource production and consumption and their associated environmental and geographical impacts” [26 p.249].
Point 5: In References, there are too few references to the latest scientific publications (from 2019, 2020 and 2021). Only found one from 2020.
Response 5: To outline potential area for future study, we add some new articles in recent years in the end of the paper, which outlined the potential reach areas for the study of AD in China.
- Hochuli,A.; Hochuli,J.; Schmid,D. Competitiveness of diversification strategies in agricultural dairy farms: Empirical findings for rural regions in Switzerland. Journal of Rural Studies 2021,82, 98–106.
- Bellon,M.R.; Kotu, B.H.; Azzarri,C.; Caracciolo, F. To diversify or not to diversify, that is the question. Pursuing agricultural development for smallholder farmers in marginal areas of Ghana. World Development 2020,125, 1-10
104.Danso-Abbeam, G.; Dagunga,G.; Ehiakpor,D. S. Rural non-farm income diversification: implications on smallholder farmers' welfare and agricultural technology adoption in Ghana. Heliyon 2020, e05393,1-11.
- Ben Fradja,N.; Jayet ,P. A.; Rozakis ,S.; Georgantad E.; Jędrejek, A. Contribution of agricultural systems to the bioeconomy in Poland: Integration of willow in the context of a stylised CAP diversification. Land Use Policy 2020,99,1-24.
106.Sharma, N.; Singh, S. P. Agricultural Diversification in Indian Punjab: An Assessment of Government Intervention Through Contract Farming. Journal of Agricultural & Food Information 2014, 15:3, 191-213, DOI: 10.1080/10496505.2014.926814
Point 6: Please adapt the References to the journal's requirements (italics, bolding of the year of publication, unnecessary spaces before commas, etc.)
Response 6: Yes, we have updated all references.
Point 7: “Please consider whether it is necessary to refer to the same source more than once with the same paragraph”. Yes, as reviewer mentioned, it is not necessary to refer to the same source more than once with the same paragraph. Therefore, we deleted [6] and [22].
Response 7: In 2007, agricultural modernization, typically characterized as “specialization, intensification and scale enlargement” [6], was first put forward by Document No.1 in China. As a feasible option for smallholder households to increase income [14], however, increased specialization and scales can endanger ecosystem functions at broader spatial scales [5]. Constrained by infeasible technologies [15, 16-17], non-specialized production is still the main mode in China’s agriculture [18].
Tackling food security in China has always been her policy priority [19], which has led to environmental deterioration arising from overuse of chemical inputs [20, 21-22]. It is imperative for policy-makers in China to balance their focus on efficient production of agricultural modernization, the maintenance of ecosystems, and prosperous rural areas pursued by AD.
Round 2
Reviewer 2 Report
Dear Authors,
I very much appreciate the efforts of the authors to meet my comments and suggestions and to implement the suggestions, observations and recommendations I made.
I am happy to inform you that I have accepted your revision of the manuscript and will recommend it for publication without further changes. Congratulations. I look forward to reading it online.
Thank you for the opportunity to let me contribute a small part to your publication.
Author Response
Thanks for your kind consideration!
Reviewer 3 Report
I think the paper has improved in the new version.
However, I think it would be of interest to point out in the conclusions the need to address some of the issues analyzed descriptively with regression models. This occurs for example in the case of the relationship between agricultural diversification and GDP.
Author Response
Changes are shown in page 17 from line 572 to page 18 line 578.
Further studies are needed from both macro and micro perspectives. Fristly from macro perspective, it is argued that there exsits an inverted U-pattern relationship between diversification and economic growth in the aggregate economy and the manufacturing sector; however, this is only partly apporved by our study on agricultural sector in China, AD index keeps relatively steady for over more than 17 years after the turning point, causes underlying the steady trend need further study.
Secondly from micro perspective,
Reviewer 4 Report
The revised manuscript may be published in the International Journal of Environmental Research and Public Health.
Author Response
Thanks for your kind consideration!